# The Role of Bacteriophage-Derived Small RNA Molecules in Bacterial and Phage Interactions

**DOI:** 10.3390/v17060834

**Published:** 2025-06-10

**Authors:** Natalia Lewandowska, Sylwia Bloch, Aleksandra Łukasiak, Wojciech Wesołowski, Grzegorz Węgrzyn, Bożena Nejman-Faleńczyk

**Affiliations:** 1Laboratory of Biology and Biotechnology of Bacteriophages, Department of Molecular Biology, Faculty of Biology, University of Gdańsk, Wita Stwosza 59, 80-308 Gdansk, Poland; natalia.lewandowska@phdstud.ug.edu.pl (N.L.); sylwia.bloch@ug.edu.pl (S.B.); aleksandra.lukasiak@phdstud.ug.edu.pl (A.Ł.); wojciech.wesolowski@phdstud.ug.edu.pl (W.W.); 2BNF—New Bio Force Ltd., Kartuska 420a, 80-125 Gdansk, Poland

**Keywords:** bacteriophage sRNAs, small regulatory RNAs, bacteriophages, microRNA

## Abstract

Small regulatory RNAs (sRNAs) play a critical role in bacterial gene expression, modulating various cellular processes, including stress responses, metabolism, virulence, and many others. While well-characterized in bacterial systems, an emerging class of phage-derived sRNAs has been identified, suggesting an underexplored regulatory network at phage–host interactions. These sRNAs, encoded within phage genomes, influence both bacterial and viral life cycles by modulating transcriptional and post-transcriptional gene expression processes. The interplay between phage-derived sRNAs and the host genome reveals a complex network of gene regulation, with an impact on bacterial fitness, pathogenesis, and horizontal gene transfer. This review explores the diverse functions of phage-encoded sRNAs, highlighting recent discoveries and their impact on bacterial physiology and phage-host interactions.

## 1. Introduction

Small regulatory RNAs (sRNAs) are essential components involved in post-transcriptional gene regulation and perform diverse functions in cellular processes [1]. In prokaryotes, these non-coding molecules typically range from 50 to 550 nucleotides in length. Their mechanisms of action include interacting with target mRNAs to influence their stability or translation efficiency, and binding to protein targets, thereby modifying the function of the bound protein. The largest group of sRNAs in prokaryotes operates through complementary base pairing with target mRNAs. Within this group, two types of molecules can be distinguished: *cis*-encoded and *trans*-encoded sRNAs (Figure 1). These two types differ in the genomic location of their coding sequences relative to their target genes [2].

*Cis*-encoded sRNAs are transcribed from the DNA strand antisense to the coding strand of the target transcript and are located within the same genomic region as their target mRNAs [2]. These sRNA molecules are characterized by long, perfect, or near-perfect complementary sequences that enable precise pairing with the target mRNAs (Figure 1A). Some authors call these molecules antisense-encoded sRNAs (asRNAs) and use this name interchangeably with *cis*-encoded sRNAs [3]. In prokaryotic organisms, asRNAs lead to the inhibition of translation or promote mRNA degradation [2]. In this work, we keep the original terms used by the authors to refer to the described sRNA molecules. In contrast, *trans*-encoded sRNAs are encoded in genomic regions distinct from their target mRNAs. They contain shorter and imperfect complementary sequences with their target mRNAs, which is why they often require chaperone proteins such as Hfq for efficient binding and stabilization (Figure 1B) [2]. *Trans*-encoded sRNAs typically do not exhibit strict one-to-one specificity. Instead, due to their short and imperfect base-pairing regions, they are capable of targeting multiple mRNA targets that share partial sequence complementarity. *Trans*-encoded sRNAs represent the most abundant class of regulatory small RNAs in prokaryotes and play roles strongly correlated to bacterial stress responses. These sRNAs act as global regulators, mediating cellular adaptation to a variety of environmental stresses, including metabolite and nutrient stress, temperature stress, envelope/outer membrane protein (OMP) stress, iron deficiency, oxidative stress, pH fluctuations, and oxygen stress or anaerobic conditions [4]. Such responses affect gene expression and induce changes in bacterial physiology and/or behavior, linking these responses to numerous other cellular processes [2].

In the 1990s, exceptionally short RNAs were identified in *Caenorhabditis elegans* and were shown to be key regulators of developmental transitions [5]. This discovery brought attention to a previously unappreciated class of sRNAs, known now as microRNAs. Found primarily in eukaryotic organisms, microRNAs are involved in post-transcriptional gene regulation and are characterized by their very short length, typically ranging from 19 to 25 nucleotides [6]. In eukaryotic cells, microRNAs are well-studied and well-defined models, in contrast to their counterparts in prokaryotic organisms [7]. Short, microRNA-size molecules have been reported in prokaryotes; however, their significant biological role remains unexplored [8]. In eukaryotic cells, microRNAs can also originate from viruses that infect them. Similarly, the first functional microRNA-size molecule, named 24B_1, derived from Shiga toxin-converting bacteriophage (shortly Stx phage) Φ24_B_ was identified a decade ago [9].

The expression of bacteriophage genes is regulated by various mechanisms, including the activity of sRNAs of phage origin. It was demonstrated that phage-derived sRNAs and prophage-carried sRNAs’ genes found within bacteriophage-derived regions of bacterial genomes can not only regulate their own gene expression but also influence the gene expression of their bacterial hosts [9,10]. A similar mechanism has been observed in eukaryotic systems, where virus-derived sRNAs can modulate host gene expression, facilitating viral infection [11,12,13]. This functional similarity between eukaryotic and prokaryotic virus-derived sRNAs suggests a conserved strategy by which viral elements influence host cellular processes.

While phage-originated sRNAs are being increasingly identified, their functions remain largely unknown and unexplored. In our previous reviews, we listed reports about phage-derived sRNAs published until that time [14,15]. In this paper, we enrich the list with new reports on identified phage-derived sRNAs and aim to explore the topic of phage-originated sRNAs, with a particular focus on their role in physiological processes within the cell. Additionally, in a separate table, we have collected 38 sRNAs without a described function (Appendix A).

## 2. Phage Replication and Lysogeny Control

While some phages (called virulent) can develop solely by multiplication in bacterial host cells, temperate bacteriophages can undergo two distinct development cycles, lytic and lysogenic. In the lytic cycle, phages hijack the bacterial machinery to produce new virions, leading to host cell lysis; while in lysogeny, the phage genome integrates into the bacterial chromosome or (rarely) occurs in the form of a plasmid, remaining unseen to bacteria until certain conditions trigger phage induction. The switch between these states is tightly regulated by complex genetic and molecular mechanisms, including sRNA of phage origin influence [9,10,14,16,17,18,19,20].

An example of phage sRNA that modulates host processes is VpdS, encoded by the lysogenic phage VP882 [16]. This sRNA plays a crucial role in promoting phage replication in *Vibrio cholerae* and functions in the quorum sensing (QS) responsive regulatory network. The VP882 phage can detect the bacterial QS molecule DPO, which triggers the switch from lysogenic to lytic development. VpdS binds to the major RNA chaperone protein Hfq, influencing both phage and host mRNAs’ levels in favor of phage replication. Specifically, it was shown to repress host-encoded sRNAs and compete with them for Hfq binding. This dynamic enables VpdS to shift the regulatory landscape to benefit phage development. Additionally, the study demonstrated that host-encoded sRNAs can antagonize phage replication by targeting phage mRNAs, suggesting a bidirectional RNA-mediated regulatory conflict between host and phage [16].

Another example of such sRNAs is PreS (LPR1), whose gene is located in λ phage genome between *ea10* and *ral* genes, and which is strongly dependent on Hfq binding [17]. PreS plays a crucial role in regulating essential bacterial genes during phage infection, serving as an adaptive strategy for λ phage to enhance its survival and replication [17]. By targeting the host *dnaN* gene, which is critical for DNA replication, PreS optimizes the cellular environment for efficient phage propagation. Notably, *dnaN* expression is upregulated under stress conditions, such as UV irradiation and exposure to alkylating agents, suggesting a potential link between PreS regulation and the bacterial SOS response. The SOS response is crucial in prophage induction as it triggers the phage developmental switch to the lytic cycle. In that study [17], two additional Hfq-dependent sRNAs of phage origin were identified—LPR2 and 6S; however, their functions were not explored (Appendix A). Nevertheless, the authors of that work speculated that those sRNAs are likely to be associated with the lytic cycle [17].

RNA 24B_1 is a small non-coding phage sRNA that plays a crucial role in maintaining the prophage state and reprogramming host metabolism. It was identified in *Escherichia coli* after induction of the Stx phage Φ24_B_ with mitomycin C [9,10]. Similar to herpesvirus microRNAs, 24B_1 is a short 20 nt molecule processed from a longer 80 nt precursor, with Hfq facilitating its maturation [9,10]. Thus, 24B_1 is a microRNA-like molecule operating in prokaryotic cells. It was reported as the first functional prokaryotic microRNA-size molecule, indicating that microRNAs may be unrestricted to eukaryotic organisms [9,10]. Overproduction of 24B_1 promotes the lysogenic developmental pathway of phage Φ24_B_ by enhancing the expression of its genes *cI, cII*, and *cIII*, which are essential for prophage maintenance [10]. The 24B_1 microRNA-size molecule is proposed to function as a negative regulator of the *d_ant* gene, which encodes a putative antirepressor protein [9,10]. By repressing *d_ant* expression, 24B_1 supports the activity of the cI repressor, thereby maintaining the repression of lytic promoters and promoting lysogeny. In the absence of 24B_1, increased *d_ant* expression leads to the inhibition of the cI repressor, resulting in enhanced expression of phage lytic genes. This shift favors more efficient lytic development, impaired lysogenization, quicker prophage induction, and the formation of partially defective virions due to the unbalanced expression of structural phage proteins [9].

An independent study identified an asRNA encoded by the phage PAK_P4, referred to as PAK_P4as15 [18]. Interestingly, such an sRNA is not encoded in the genome of a related phage PAK_P3. This asRNA is associated with the endolysin gene, and it may act as a translational repressor, potentially delaying the lysis of the host cell [18]. Four other sRNAs (PAK_P4as06, PAK_P4as07, PAK_P4as14, and PAK_P4as16) have been identified in this phage, but their functions have not been determined [18].

The small sar asRNA plays a crucial role in the regulation of gene expression in bacteriophage P22, particularly in the control of antirepressor synthesis [19]. The key function of sar asRNA is repression of the antirepressor by base pairing with the *ant* mRNA, which encodes the antirepressor protein. This base pairing interferes with the initiation of translation of the antirepressor, effectively reducing its synthesis at a post-transcriptional level. Despite the presence of other repressors of *ant* gene expression, sar asRNA is essential for the proper regulation of antirepressor synthesis, which is critical for the establishment of lysogeny in P22. The presence of sar asRNA ensures optimal antirepressor levels, maintaining the proper balance of gene expression throughout the phage cycles [19].

OOP RNA is a small regulatory RNA encoded by lambdoid bacteriophages. It is transcribed from the *p*O promoter and was initially believed to play a dual role in both the expression of the *cI* gene and as a primer for DNA replication [15]. However, later studies clarified that while OOP RNA does not serve as a primer for DNA synthesis, the *p*O promoter remains important for regulating DNA replication. One of the key functions of the OOP RNA is its role as an asRNA for the *cII* gene, which encodes a crucial regulator of phage development, named cII. By binding to *cII* mRNA, OOP RNA inhibits its expression and thereby influences whether the phage undergoes lytic or lysogenic development [15]. Additionally, OOP RNA expression can suppress the Rex exclusion phenomenon, which typically prevents the development of certain bacteriophages that infect lysogenic bacteria. On the transcriptional level, OOP RNA promotes the degradation of *cII* mRNA through RNase III activity, and its overexpression tends to favor the lytic cycle of bacteriophage λ over lysogeny. Recent research has also uncovered additional small RNAs encoded by lambdoid bacteriophages, suggesting that the regulatory roles of RNAs like OOP RNA are even more complex and significant than previously understood [15]. Overall, OOP RNA plays a critical role in regulating gene expression of phage origin and shaping the developmental pathways of bacterial viruses [15].

The aQ RNA is an asRNA produced from the *p*aQ promoter during the development of phage λ and plays a key role in the regulation of gene expression [20]. Specifically, aQ RNA downregulates the expression of the *Q* gene, whose product is essential for the activation of late genes during the phage’s lytic development. aQ asRNA is proposed to act *in cis* to regulate *Q* gene expression. Additionally, aQ RNA appears to function alongside other promoters, such as *p*RE and *p*I, to coordinate the lysogenic growth response, suggesting its involvement in a broader regulatory network. By controlling *Q* gene expression, aQ RNA critically influences the decision between lysogenic and lytic cycles, thereby shaping the phage developmental pathway [20]. Overall, aQ RNA exemplifies the sophisticated regulatory strategies employed by phages to fine-tune gene expression and balance their life cycles [20].

## 3. Bacterial Cell Division and Morphology

Phage sRNAs enable viruses to manipulate host physiology to optimize infection success or persistence, often by targeting key regulatory pathways [14,21,22,23,24,25]. One such example is DicF sRNA from the Qin prophage. This small molecule regulates growth, cell division, and metabolic functions, and influences adaptation to environmental conditions of *E. coli* bacteria [21,22,23,24,25]. DicF (53 nt) is transcribed from the *dicB* operon and functions as a *trans*-acting inhibitor of cell division, particularly under stress conditions such as elevated temperatures [21,23,24,25]. This sRNA inhibits bacterial cell division by directly base-pairing with *ftsZ* mRNA, preventing translation and blocking the synthesis of FtsZ, a bacterial tubulin homolog, essential for cytokinesis [21,22,24,25]. In addition to affecting *ftsZ*, DicF also regulates metabolic genes, and its production can be toxic to *E. coli* [21]. It has been shown that DicF represses *xylR* and *pykA* mRNAs, which are involved in xylose metabolism and pyruvate kinase activity, respectively [21]. The influence of DicF on bacterial morphology is particularly clear under anaerobic conditions, where it promotes cell filamentation by suppressing FtsZ production [21,22,25]. This regulation is stabilized by the interaction of DicF with enolase, a component of the RNA degradation system, which allows the sRNA to accumulate and exert its inhibitory effect. The resulting filamentous growth pattern is especially relevant in pathogenic strains of *E. coli*, where multiple copies of DicF facilitate morphological adaptations in oxygen-limited environments [25]. The Hfq protein further increases DicF stability, protecting it from degradation and enhancing its ability to inhibit FtsZ [21,25]. This regulatory mechanism enables *E. coli* to adjust its morphology in response to fluctuating oxygen levels, highlighting DicF’s role in bacterial stress adaptation. The inhibitory effects of DicF extend beyond *ftsZ* repression, as its secondary structure resembles a Rho-independent transcription terminator, suggesting that it may modulate RNA stability and degradation of division-related transcripts [21,25].

In summary, as an untranslated product of the *dicB* operon, DicF exemplifies the sophisticated post-transcriptional regulation empowered by small RNAs to fine-tune bacterial cell cycle control. In addition to its role in cell division, DicF also regulates metabolic processes.

## 4. Superinfection Exclusion

Superinfection exclusion (SIE) is a strategy employed by bacteriophages to prevent further infections of the same host cell by other phages. This is achieved through various mechanisms, such as modifying bacterial surface receptors to block phage adsorption or inhibiting the injection of phage genetic material into the host cell. Although SIE provides short-term benefits to phages by monopolizing host cells, it may cause long-term costs, such as reduced burst size [26].

STnc6030 is a phage-derived asRNA transcribed from the BTP1 prophage genome and has been proposed to regulate gene expression by base-pairing with complementary mRNA sequences. Specifically, STnc6030 interacts with the mRNA of the BTP1 tailspike gene, reducing transcript stability and inhibiting translation [27]. This post-transcriptional regulation not only modulates expression of the tailspike protein—a key factor in host recognition and DNA injection—but also plays a pivotal role in superinfection exclusion, a mechanism that enables lysogenized bacteria to resist reinfection by the same or closely related phages. This exclusion contributes to the maintenance of the prophage state by preventing excessive phage replication and promoting the genomic stability of the host. Supporting this model, escape mutants of BTP1 were identified that could overcome the exclusion conferred by STnc6030. Sequencing of these variants revealed single nucleotide polymorphisms clustered in the region antisense to the tailspike gene, highlighting the importance of specific base-pairing interactions in the regulatory mechanism [27]. Notably, while STnc6030 limits superinfection, it does not impair induction of the BTP1 prophage, suggesting a fine-tuned regulatory system that selectively restricts external infection without interfering with prophage activation [27].

Prophage-encoded ncRNAs also mediate non-virulence accessory functions, including the sas asRNA of phage P22 that induces a translational switch between distinct peptides encoded by the *sieB* gene and is critical to the function of the SieB superinfection-exclusion system [28]. The mechanism by which sas contributes to SIE involves its function as an asRNA that regulates the expression of the *sieB* gene, which encodes the SieB exclusion protein, that is known to inhibit lytic development of superinfecting phages such as P22 [28]. asRNA sas exerts its effect through base-pairing interactions with *sieB* mRNA, modulating its translation. Disruption of this RNA–RNA interaction, for instance by targeted mutations, leads to altered *sieB* expression and consequently affects superinfection efficiency. This suggests that precise base-pairing between sas and its target mRNA is a key element of the exclusion mechanism [28]. Additionally, the system displays a functional redundancy or overlap: even if *sieB* regulation is impaired, the Esc protein, which can be expressed by either the resident prophage or the incoming phage, can help bypass SieB-mediated exclusion. This indicates a complex regulatory balance between sas, SieB, and Esc.

## 5. Virulence and Stress Response

Bacteriophages play significant roles in modulating bacterial virulence and stress responses. When infecting bacteria, phages can trigger stress responses by introducing DNA damage or interfering with bacterial metabolic processes. In turn, bacterial cells activate defense mechanisms like the SOS response to cope with the phage infection. Additionally, some bacteriophages may influence bacterial virulence by carrying virulence factors or regulatory genes, which can enhance the pathogenicity of the host bacteria. This interaction between phages and bacteria shapes not only the bacterial stress response but also the potential for later disease transmission to humans [29].

For example, StxS functions as a regulatory sRNA that influences both Shiga toxin 1 production and the general stress response in *E. coli* O157:H7 [30]. It directly interacts with the *stx1AB* transcript, repressing Shiga toxin 1 production under lysogenic conditions. Additionally, StxS sRNA activates RpoS, the stationary-phase sigma factor, by binding to an activating seed sequence within the 5′ UTR, a mechanism typically utilized by host-encoded sRNAs. This activation enhances bacterial growth at high cell densities under nutrient-limiting conditions, demonstrating how RNAs derived from phage regulation can obtain regulatory functions that impact host fitness [30].

The prophage-encoded small RNA SprY is another example of phage sRNAs, and it plays a crucial role in regulating virulence gene expression in *Staphylococcus aureus* [31]. SprY binds to RNAIII, a key virulence regulator, specifically targeting its 13th stem-loop, which is responsible for repressing multiple mRNA targets. By interacting with RNAIII, SprY increases the expression of certain virulence repressors while reducing hemolytic activity, ultimately decreasing *S. aureus* pathogenicity through an RNA sponge-like activity. mRNAs encoding the repressor of the Rot toxin and the extracellular complement-binding protein Ecb are among the targets whose expression is increased by SprY binding to RNAIII [31].

## 6. Host Metabolism and Environmental Adaptation

Bacteriophages rely on their host’s cellular machinery by hijacking bacterial molecular systems for phage replication, transcription, and translation [32,33]. Phage-derived regulatory sRNAs play roles in controlling these processes, influencing bacterial metabolism while also modulating phage infection.

One such example is the 24B_1 microRNA-size molecule identified in *E. coli* after induction of Φ24_B_ prophage [9,10]. 24B_1 alters the host’s central carbon metabolism by binding to the *sdhB* mRNA and reducing intracellular ATP levels. Such a metabolic shift resembles the Warburg effect observed in cancer cells [11].

ipeX is another small RNA that plays a significant role in regulating gene expression in bacteria, particularly in the context of phage infection [34]. ipeX is known to regulate the expression of transcripts encoded by the core genome of the host bacteria and is synthesized after transcription from the phage’s porin gene promoter, as part of the phage porin transcript. Specifically, ipeX has been shown to control the production of OmpC, a porin in *E. coli*, which is crucial for the bacterial outer membrane’s permeability and function. The regulation by ipeX occurs through a mechanism that does not rely only on sequence complementarity with *ompC* mRNA. This suggests that additional factors may be involved in its regulatory function. The production of ipeX itself requires processing from a larger precursor transcript, a characteristic shared with eukaryotic microRNA [34,35].

Four sRNAs (abiF, wcaG, manA, glnA) were identified in the S-CREM1 cyanophage genome [36,37,38]. The abiF sRNA shares a conserved motif with *abiF* genes from various bacteria and bacteriophages, suggesting it may play a functional role during phage infection, though its exact regulatory role remains unconfirmed (Appendix A). Additionally, three *cis*-regulatory sRNAs, wcaG, manA, and glnA, with conserved motifs, were also found in other cyanophages. The wcaG sRNA is likely involved in regulating genes related to exopolysaccharide production, while the manA sRNA is associated with genes involved in nucleotide synthesis, carbohydrate metabolism, and photosynthesis. Lastly, the glnA sRNA is predicted to regulate genes related to nitrogen metabolism, including those encoding nitrogen regulatory proteins and transporters [36,37,38].

φR1-37 is a bacteriophage that infects *Yersinia enterocolitica*, and its RNA plays a pivotal role during lytic infection, impacting both phage and host gene expression [39]. Ten novel RNA species were identified as being transcribed from the phage genome [39]. Nine of these (misc_1-3 and misc_5-10) originate from the antisense strand, suggesting they function as asRNAs, but their exact role is unknown (Appendix A), while one, called misc_4, is encoded within an intragenic region and may interact with host mRNAs [39]. Bioinformatic analysis using TargetRNA2 predicted three potential host targets for misc_4: *ptr*, *tuf*A, and *ddrA* that encode protease III, factor Tu, and the reactivation protein for propanediol utilization diol dehydratase, respectively [39].

Another study identified an asRNA linked to the *psbA* gene in the cyanophage S-PM2 [40]. This asRNA is designated as Cyanophage Functional RNA I (CfrI) and is encoded in the intergenic region between *psbA* and the gene coding for the homing endonuclease F-ChpI. This designation highlights its unique role in the context of cyanophage biology. One of the proposed functions of this asRNA is to prevent the formation of mobile group I introns within the *psbA* gene of cyanophages. This could be crucial for maintaining the integrity of the *psbA* gene during the phage’s lifecycle. The presence of a gene encoding this asRNA at the 3′ end of the *psbA* gene in various cyanophages implies that such a regulatory mechanism may be evolutionarily conserved [40].

Bacterial motility may contribute to virulence or improve bacterial survival in various environments. Esr41 is a small RNA from an intergenic region unique to the enterohemorrhagic *E. coli* (EHEC) O157:H7 Sakai strain, which is absent in the genome of the nonpathogenic *E. coli* K-12 [41]. The sRNA enhances the expression of the *fliC* gene, leading to increased bacterial cell motility [41]. Motility assays showed that overexpression of Esr41 from a multicopy plasmid significantly increased the motility of EHEC. This is linked to the upregulation of *fliC*, which encodes the major flagellar filament protein, FliC. As a *trans*-acting, Hfq-binding sRNA, Esr41 was proposed to regulate motility-related genes through base-pairing interactions with target mRNAs, particularly within the 5′ untranslated region [41]. Its role in motility enhancement is conserved across different *E. coli* strains, including the non-pathogenic *E. coli* K-12, suggesting a broader regulatory function. Additionally, Esr41 likely participates in a complex motility regulation network, interacting with other sRNAs and transcription factors to coordinate flagellar gene expression in response to environmental changes, highlighting its potential role in bacterial adaptation and pathogenicity [41].

## 7. Regulation of Gene Expression

Bacteriophages, as viruses, may manipulate key pathways in their bacterial hosts at the transcriptional level, optimizing conditions for their replication. By influencing gene expression, phages create optimal conditions that favor their propagation [42].

IsrK sRNA, encoded by the Gifsy-1 prophage, functions as both a small RNA and a messenger RNA, modulating key bacterial processes [43]. It regulates gene expression by activating *antQ*, a gene encoding an antiterminator protein AntQ, located upstream of *isrK*. Raised levels of IsrK sRNA have been shown to inhibit *Salmonella* growth, an effect linked to the expression of *antQ*. The protein disrupts transcription termination, creating conflicts between transcription and replication that lead to DNA damage. When AntQ accumulates at high levels, bacterial growth is suppressed, ultimately resulting in cell death [43]. Then, stress occurs within the bacterial cell and ultimately results in a growth decrease and leads to cell death. Additionally, AntQ promotes the formation of R-loops, hybrid RNA–DNA structures that can initiate DNA replication independently of the origin of replication (*oriC*). This uncontrolled replication leads to DNA damage, further contributing to its cytotoxic effects. Bacterial cells expressing *antQ* also exhibit condensed chromatin morphology, suggesting a broader impact on genome organization. Despite these harmful effects, bacterial cells have mechanisms to counteract AntQ toxicity [43]. The co-expression of transcription termination factors such as Rho and RNase H helps restore normal transcription processes, resolving R-loops, and maintaining genomic integrity, thereby mitigating AntQ-induced stress. Experimental data showed that a*ntQ* expression significantly reduced bacterial survival rates [43]. Together, these findings highlight AntQ as a potent regulator of bacterial physiology, capable of disrupting transcription, damaging DNA, and severely impacting cell survival, whose action is controlled by sRNA [43].

Moreover, the IsrK sRNA plays a crucial role in modulating gene expression, particularly in the regulation of open reading frame *orf45* and *anrP* gene. IsrK is essential for the activation of the *anrP* gene, which is located downstream of *orf45*. This activation occurs through a mechanism involving the translation of *orf45*, suggesting that IsrK enhances downstream gene expression by facilitating *orf45* translation. In other words, the regulation of *anrP* gene by IsrK occurs by translational coupling, in which the translation of *orf45* is a necessary point for *anrP* expression. In effect, IsrK promotes the translation of *orf45*, thereby enabling the later translation of gene *anrP*. Comparative sequence analysis indicates that while the nucleotide sequence of *orf45* is conserved among the *Enterobacteriaceae* family, its amino acid sequence exhibits variability. This suggests that the regulatory interaction between *orf45* and IsrK may be an important feature conserved across different bacterial species [43].

Roles of small regulatory RNAs that are derived from bacteriophages were discussed recently [44,45], particularly focusing on their roles in the regulation of gene expression in EHEC. AgvB sRNA is encoded by the Sp5 prophage of enterohemorrhagic *E. coli*. Unlike typical sRNAs, it does not function as an antisense regulator but instead inhibits the activity of another sRNA, GcvB, encoded in the *E. coli* chromosome. By mimicking the mRNA targets of GcvB, AgvB counteracts its regulatory effects, allowing the expression of genes otherwise repressed by GcvB. This global regulator controls the translation of up to 1% of transcripts [44]. AgvB relieves the translational repression of the dipeptide transporter DppA, which GcvB normally represses. This indicates that AgvB plays a crucial role in regulating amino acid transport and metabolism in enterohaemorrhagic *E. coli* [44,45].

Another identified sRNA is AsxR from the Shiga-toxin-2-encoding phage of *E. coli* O157:H7 Sakai strain. This molecule functions as an asRNA that increases the translation of ChuS, a heme oxygenase, by destabilizing the small regulatory FnrS RNA [44]. FnrS typically represses ChuS translation by binding to the Shine–Dalgarno sequence, but AsxR disrupts FnrS stability by binding to its Rho-independent terminator stem-loop. This interaction reduces FnrS levels and its association with the Hfq chaperone, thereby soothing the repression of ChuS translation [44].

The c4 RNA is a regulatory asRNA found in bacteriophages P1 and P7, playing a critical role in controlling gene expression by targeting *ant* mRNA, which encodes antirepressor protein [46]. Its primary function is to inhibit the synthesis of antirepressors. This mechanism is essential for the heteroimmunity observed between P1 and P7, allowing them to prevent co-infection of the same bacterial host. c4 RNA operates through sequence-specific binding to complementary regions of the *ant* mRNA, particularly around the ribosome binding site, thereby blocking translation [46]. The function of c4 RNA is strongly dependent on its secondary structure, with the complementary sequences likely positioned within RNA loops to facilitate efficient binding. Research into c4 RNA has also highlighted its potential as a prototype of a new class of regulatory RNAs, suggesting that additional, similarly functioning asRNAs might exist. Overall, c4 RNA is a key regulatory element that modulates phage immunity and gene expression through sophisticated antisense interactions [46].

## 8. Concluding Remarks

Bacteriophage-derived small RNAs represent a newly emerging layer of genetic regulation in bacteria, influencing essential cellular functions and the complex interactions between bacteriophages and their hosts (Figure 2). These sRNAs contribute to the regulation of bacterial metabolism, stress response, and virulence by modulating gene expression, often through mechanisms that resemble eukaryotic virus sRNAs [35]. Their regulatory functions can significantly impact bacterial physiology, shaping the outcomes of phage infection and integration. A functional classification of phage-derived small RNAs, based on cellular processes they modulate, is presented in Table 1.

One key role of phage-derived sRNAs is their ability to modify host metabolism, which resembles microRNA in herpesviruses in eukaryotic cells [11]. For instance, the 24B_1 sRNA from bacteriophage Φ24_B_ influences central carbon metabolism by interacting with *sdhB* mRNA, ultimately reducing ATP levels within the bacterial cell [10]. Beyond metabolic regulation, these sRNAs are also implicated in stress responses. While their precise functions are not always fully explained, the well-established role of *trans*-encoded sRNAs in bacterial stress adaptation suggests that some phage-derived sRNAs may similarly contribute to host resilience under adverse conditions [4].

Virulence regulation is another crucial aspect of phage sRNAs’ activity. In *S. aureus*, the prophage-encoded SprY sRNA modulates the expression of virulence factors by binding to RNAIII, a central regulatory RNA involved in pathogenesis [31]. Similarly, the phage-derived Esr41 sRNA enhances bacterial motility in enterohemorrhagic *E. coli* [41]. These examples highlight how phage sRNAs can actively shape the interaction between bacteria and their environment, influencing host adaptability and infectivity.

The molecular mechanisms employed by phage-derived sRNAs are diverse, often mirroring those of bacterial regulatory RNAs. Many such regulatory molecules function as asRNAs, pairing with complementary mRNA sequences to regulate translation and stability. A notable example is PAK_P4as15, an asRNA from phage PAK_P4 that may repress endolysin translation, potentially delaying bacterial cell lysis [18]. Additionally, some sRNAs interact with RNA-binding proteins, such as the chaperone Hfq, which stabilizes sRNAs and eases their regulatory interactions. For instance, the phage-derived VpdS [16] and PreS [17] sRNAs require Hfq for efficient mRNA binding, while AgvB competes with bacterial sRNAs for Hfq binding, interfering with host regulatory pathways [44].

A particularly interesting aspect of phage sRNAs is their resemblance to eukaryotic microRNAs, both in size and maturation (from original transcripts) and function, like 24B_1 sRNA. Not only is its action similar to microRNAs of herpesviruses but also 24B_1 is processed from a longer 80 nt precursor that resembles the biogenesis of eukaryotic microRNAs [9,10,11]. Although bacteria lack the complex RNA maturation system found in eukaryotes [35], the Hfq protein was demonstrated to be involved in the maturation process [10].

Beyond their roles in modulating host physiology, phage-derived sRNAs significantly impact the bacteriophage life cycle by regulating the lysogenic–lytic switch. These molecules help to determine whether a phage remains integrated within the host genome or initiates induction and replication. For example, 24B_1 sRNA promotes lysogeny in phage Φ24_B_ [9,10]. VpdS, encoded by the temperate phage VP882, plays a key role in triggering phage replication in *V. cholerae* in response to quorum-sensing signals [16]. Additionally, certain phage sRNAs optimize conditions for their replication by targeting bacterial genes critical for DNA synthesis, such as the *dnaN* gene targeted by PreS in λ phage [17]. Other sRNAs, like PAK_P4as15, may extend the time frame for viral replication by delaying host cell lysis [18].

The similarities between phage-derived sRNAs and eukaryotic microRNAs extend beyond function to evolutionary implications. The presence of mobile genetic elements, such as bacteriophages, could have facilitated the spread and improvement of these regulatory mechanisms. Additionally, while the Hfq protein in bacteria is not homologous to eukaryotic Argonaute proteins, it serves a similar function in presenting sRNAs to their targets, further empowering the functional parallels between these systems [35].

Bacteriophage-derived sRNAs represent an important but still largely unexplored field of bacterial and phage regulation, influencing host metabolism, stress adaptation, virulence, and the phage cycles. Their functional similarities to eukaryotic microRNAs highlight the unconventional view that microRNA-based gene regulation is not exclusive to eukaryotes. Thus, studying phage sRNAs could provide key insights into microbial genetics and lead to new applications in biotechnology, medicine, and treatments such as phage therapy.

The findings discussed in this review article illustrate the complex interactions between phage-derived sRNAs and their host bacteria, emphasizing their potential impact on bacterial behavior and pathogenicity. The structural and functional parallels between phage sRNAs and eukaryotic microRNAs highlight how phage-derived sRNAs can shape bacterial transcriptional networks in a way that eukaryotic viruses do in their hosts.

## Figures and Tables

**Figure 1 viruses-17-00834-f001:**
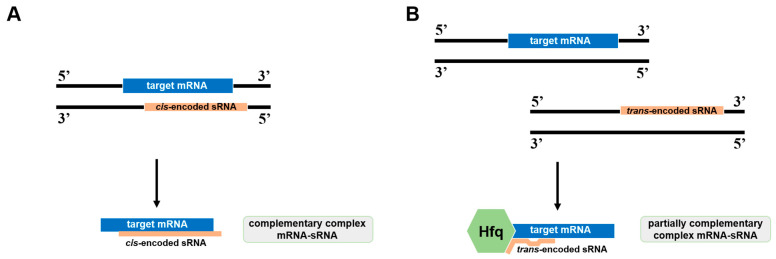
Difference in the genomic location of *cis-*ecoded (**A**) and *trans*-encoded (**B**) sRNAs to their target gene. Orange boxes show the sRNA, while blue boxes show the target mRNA. Hfq protein is marked in green.

**Figure 2 viruses-17-00834-f002:**
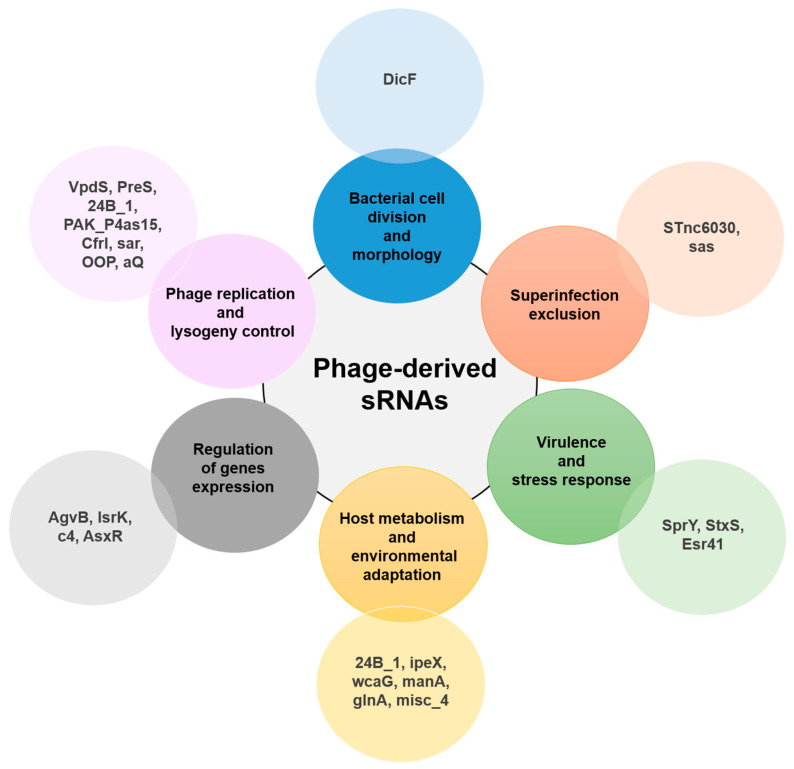
Roles of phage-encoded sRNAs in different cellular processes: The experimentally validated sRNAs are depicted along with the respective biological processes.

**Table 1 viruses-17-00834-t001:** Functional classification of phage-derived small RNAs based on cellular processes.

Cellular Process	sRNA Name	Function	Host	Phage	Size (nt)	sRNA Acting Type	Reference
**Phage Replication and Lysogeny Control**	VpdS	Regulates phage replication in *V. cholerae*, linking quorum sensing to phage life cycle.	*Vibrio cholerae*	VP882	~50	*trans*	[16]
	PreS (LPR1)	Regulates the expression of *dnaN* gene, optimizing conditions for λ phage propagation.	*Escherichia coli*	λ	~88	*trans*	[17]
	24B_1	Promotes lysogeny by enhancing the expression of *cI*, *cII*, and *cIII* genes and altering host metabolism.	*Escherichia coli*	Φ24_B_	20	*trans*	[9,10]
	PAK_P4as15	Acts as a translational repressor of the endolysin gene, delaying host cell lysis.	*Pseudomonas aeruginosa*	PAK_P4	~550	*cis*	[18]
	CfrI	Prevents mobile intron formation in *psbA* gene of cyanophages, ensuring stability.	*Synechococcus* sp. WH7803	S-PM2	~225	*cis*	[38]
	sar	Pairs with *ant* mRNA and inhibits antirepressor synthesis.	*Salmonella enterica* serovar Typhimurium	P22	68/69	*trans*	[19]
	OOP	Controls *cII* gene expression and influences the lysis–lysogeny.	*Escherichia coli*	λ and Stx phages	77	*cis*	[15]
	aQ	Regulates *Q* gene expression	*Escherichia coli*	λ	220	*cis*	[20]
**Bacterial Cell Division and Morphology**	DicF	Inhibits expression of *ftsZ* gene and cell division—promotes cell filamentation under anaerobic conditions.	*Escherichia coli*	Qin	53	*trans*	[21,22,23,24,25]
**Superinfection Exclusion**	STnc6030	Mediates superinfection exclusion of phage BTP1.	*Salmonella enterica* serovar Typhimurium D23580	BTP1	786	*cis*	[27]
	sas	Induces a translational switch in *sieB*, crucial for the SieB superinfection-exclusion system.	*Salmonella enterica* serovar Typhimurium	P22	105	*cis*	[28]
**Virulence and Stress Response**	SprY	Regulates RNAIII activity, reducing the virulence and hemolytic activity of the bacterial cell.	*Staphylococcus aureus*	prophage φ12	∼125	*trans*	[31]
	StxS	Represses Shiga toxin 1 production and activates *rpoS*, enhancing stress resistance in *E. coli* O157:H7.	*Escherichia coli* O157:H7	StxΦ	74	*trans*	[30]
	Esr41	Stimulates *fliC* expression, increasing bacterial motility, which may influence virulence.	*Escherichia coli* O157:H7 str. Sakai	Sakai prophage- like element 1 (SpLE1)	~70	*trans*	[39]
**Host Metabolism and Environmental Adaptation**	24B_1	Alters carbon metabolism by binding to *sdhB* mRNA, reducing ATP levels.	*Escherichia coli*	Φ24_B_	20	*trans*	[9,10]
	ipeX	Regulates the expression of *ompC* gene, affecting membrane permeability.	*Escherichia coli*	cryptic prophage DLP12	167	*cis*	[34]
	wcaG	Probably regulates the expression of genes linked to exopolysaccharide production.	*Synechococcus*	S-CREM1	97	*cis*	[36,37,38]
	manA	Associated with nucleotide synthesis, carbohydrate metabolism, and photosynthesis.	*Synechococcus*	S-CREM1	214	*cis*	[36]
	glnA	Probably regulates the expression of genes related to nitrogen metabolism.	*Synechococcus*	S-CREM1	98	*cis*	[36]
	misc_4	Predicted to interact with host mRNAs, including *ptr*, *tufA,* and *ddrA*.	*Yersinia enterocolitica*	φR1-37	296	*trans*	[37]
**Regulation of Gene Expression**	AgvB (EcOnc01)	Counteracts the regulatory effects of the bacterial GcvB sRNA by mimicking its mRNA targets.	*Escherichia coli* O157:H7 str. Sakai	lambdoid prophage Sp5	~60	*trans*	[44,45]
	IsrK	Regulates transcription termination through *antQ*, affecting *Salmonella* growth. Has a dual role as sRNA and mRNA.	*Salmonella typhimurium*	prophage Gifsy-1	217	*cis* and *trans*	[43]
	c4	Controls gene expression by targeting *ant* mRNA, which encodes antirepressor proteins.	*Escherichia coli*	P1 and P7	77	*trans*	[46]
	EcOnc02 (AsxR)	Derepresses a heme oxygenase. Enhances ChuS translation by destabilizing the FnrS sRNA.	*Escherichia coli* O157:H7 str. Sakai	lambdoid prophage Sp5	~54	*trans*	[44]

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
