# Peer review of "The Role of Bacteriophage-Derived Small RNA Molecules in Bacterial and Phage Interactions"

_viruses, 2025, doi:10.3390/v17060834_

Round 1

Reviewer 1 Report

Comments and Suggestions for Authors

The work entitled "The role of bacteriophage-derived small RNA molecules in bacterial and phage interactions" by Lewandowska et al. review recent discoveries on the diverse functions of phage-encoded sRNAs and their impact on bacterial physiology and phage-host interactions. Overall, this is a well-written paper and gives us a comprehensive overview of the types and functions of phage-encoded sRNAs. I have several concerns on this manuscript.

  1. Lines 60-67, the author did not introduce the mode of trans sRNA interacting with their target. Is it one-to-one specific or the trans sRNA can have an effect with any mRNA that can coarsely match with it?
  2. Lines 105-112, the introduction of the VpdS function is not clear. The authors did introduce what the targets of the VpdS sRNA, how the VpdS function links to the phage response to the QS molecule.
  3. Lines 122-124, Since the function of the two sRNAs are not explored, how the authors infer the function of these two sRNAs are associated with lytic cycle?
  4. Lines 149-147, It is more meaningful to drag this section to the subheading of “6. Host Metabolism and Environmental Adaption”
  5. Lines 202-205, Please rephrase those sentences. Delete the one in Lines 204-205, and put the information of the RNA length (53-nt) in the sentence in 202-203.
  6. Lines 208-209, this sentence is repeated with those in Lines 222-226.
  7. Lines 237-246, the authors did not introduce the way that how STnc6030 have functions in the Superinfection Exclusion
  8. Lines 274-289, It is more meaningful to move this part into the subheading of “6. Host Metabolism and Environmental Adaption”
  9. Lines 309-318, the references listed for this part are not accurate. The detailed references for the function studies on those sRNAs should be listed.

Author Response

Dear Reviewer,

Thank you very much for your helpful and insightful comments. We greatly appreciate your suggestions and have carefully considered them in our revisions. All fragments changed by us in the manuscript are marked in yellow and, for convenience, also quoted below.

COMMENTS 1 -  Lines 60-67, the author did not introduce the mode of trans sRNA interacting with their target. Is it one-to-one specific or the trans sRNA can have an effect with any mRNA that can coarsely match with it?

RESPONSE 1 - Thank you for this remark. The information has been added in lines 60-63.

“Trans-encoded sRNAs typically do not exhibit strict one-to-one specificity. Instead, due to their short and imperfect base-pairing regions, they are capable of targeting multiple mRNA targets that share partial sequence complementarity.”

COMMENT 2 - Lines 105-112, the introduction of the VpdS function is not clear. The authors did introduce what the targets of the VpdS sRNA, how the VpdS function links to the phage response to the QS molecule.

RESPONSE 2 - Text was clarified and updated. Lines 108-118

“An example of phage sRNA that modulates host processes is VpdS, encoded by the lysogenic phage VP882 [16]. This sRNA plays a crucial role in promoting phage replication in Vibrio cholerae and functions as a quorum sensing (QS) responsive regulatory network. The VP882 phage can detect the bacterial QS molecule DPO, which triggers the switch from lysogenic and lytic development. VpdS binds to the major RNA chaperone protein Hfq, influencing both phage and host mRNAs’ levels in favor of phage replication. Specifically, it was shown to repress host-encoded sRNAs and compete with them for Hfq binding. This dynamic enables VpdS to shift the regulatory landscape to benefit phage development. Additionally, the study demonstrated that host-encoded sRNAs can antagonize phage replication by targeting phage mRNAs, suggesting a bidirectional RNA-mediated regulatory conflict between host and phage [16].”

COMMENT 3 - Lines 122-124, Since the function of the two sRNAs are not explored, how the authors infer the function of these two sRNAs are associated with lytic cycle?

RESPONSE 3 - The authors of the original manuscript assumed that in the paper. The sentence was rearranged to indicate that. Lines 130-131.

“Nevertheless, the authors of that work speculated that those sRNAs are likely to be associated with the lytic cycle.”

COMMENT 4. Lines 149-147, It is more meaningful to drag this section to the subheading of “6. Host Metabolism and Environmental Adaption”

RESPONSE 4 - The entire paragraph about CrfI asRNA has been moved following the reviewer’s suggestion. Lines 325-333   

COMMENT 5 -  Lines 202-205, Please rephrase those sentences. Delete the one in Lines 204-205, and put the information of the RNA length (53-nt) in the sentence in 202-203.

RESPONSE 5 - The sentences were rephrased. Lines 199-201

“DicF (53-nt) is transcribed from the dicB operon and functions as a trans-acting inhibitor of cell division, particularly under stress conditions such as elevated temperatures [21, 23-25].”

COMMENT 6 - Lines 208-209, this sentence is repeated with those in Lines 222-226.

RESPONSE 6 - Since the first reviewer also had some comments on this fragment, we have modified it to reconcile both suggestions. We hope that it is better now. Lines 199-206

“DicF (53-nt) is transcribed from the dicB operon and functions as a trans-acting inhibitor of cell division, particularly under stress conditions such as elevated temperatures [21, 23-25]. This sRNA inhibits bacterial cell division by directly base-pairing with ftsZ mRNA, preventing translation and blocking the synthesis of FtsZ, a bacterial tubulin homolog, essential for cytokinesis [21, 22, 24, 25]. Aside from affecting ftsZ, DicF also regulates metabolic genes, and its production can be toxic to E. coli [21]. It has been shown that DicF represses xylR and pykA mRNAs, which are involved in xylose metabolism and pyruvate kinase activity, respectively [21].”

COMMENT 7 -  Lines 237-246, the authors did not introduce the way that how STnc6030 have functions in the Superinfection Exclusion

RESPONSE 7 - The fragment was supplemented with this information. Lines 230-245. We are sorry with the color, but this one is marked in green as the first reviewer also asked about it.

“STnc6030 is a phage-derived antisense RNA (asRNA) transcribed from the BTP1 prophage genome and has been proposed to regulate gene expression by base-pairing with complementary mRNA sequences. Specifically, STnc6030 interacts with the mRNA of the BTP1 tailspike gene, reducing transcript stability and inhibiting translation [27]. This post-transcriptional regulation not only modulates expression of the tailspike protein - a key factor in host recognition and DNA injection - but also plays a pivotal role in superinfection exclusion, a mechanism that enables lysogenized bacteria to resist reinfection by the same or closely related phages. This exclusion contributes to the maintenance of the prophage state by preventing excessive phage replication and promoting the genomic stability of the host. Supporting this model, escape mutants of BTP1 were identified that could overcome the exclusion conferred by STnc6030. Se-quencing of these variants revealed single nucleotide polymorphisms clustered in the region antisense to the tailspike gene, highlighting the importance of specific base-pairing interactions in the regulatory mechanism [27]. Notably, while STnc6030 limits superinfection, it does not impair induction of the BTP1 prophage, suggesting a fine-tuned regulatory system that selectively restricts external infection without interfering with prophage activation [27].”

COMMENT 8 - Lines 274-289, It is more meaningful to move this part into the subheading of “6. Host Metabolism and Environmental Adaption”

RESPONSE 8 - It was moved following the suggestion of the reviewer. Lines 334-348

COMMENT 9 - Lines 309-318, the references listed for this part are not accurate. The detailed references for the function studies on those sRNAs should be listed.

RESPONSE 9 - We apologise for the oversight. The two references have been added to the list and cited in the text.

 [38] Weinberg Z, Wang JX, Bogue J, Yang J, Corbino K, Moy RH, Breaker RR. Comparative genomics reveals 104 candidate structured RNAs from bacteria, archaea, and their metagenomes. Genome Biol. 2010;11(3):R31. doi: 10.1186/gb-2010-11-3-r31.

 [39] Wang Q, Cai L, Zhang R, Wei S, Li F, Liu Y, Xu Y. A Unique Set of Auxiliary Metabolic Genes Found in an Isolated Cyanophage Sheds New Light on Marine Phage-Host Interactions. Microbiol Spectr. 2022 Oct 26;10(5):e0236722. doi: 10.1128/spectrum.02367-22. 

Reviewer 2 Report

Comments and Suggestions for Authors

The manuscript summarizes the knowledge about regulatory RNAs of phage origin. The main text describes in detail properties of these molecules according to the published papers. Sometimes the text is too detailed and without schemes it is hard to follow. However, in general the manuscript is well written and do not contain substantial mistakes.

Specific comments:

Line 225: The sentence „It has been shown that DicF represses xylR and pykA mRNAs, which are involved in xylose metabolism and pyruvate kinase activity, respectively [22].“ should be moved into the line 209.

Line 240: What is the mechanism by which STnc6030 contributes to superinfection exclusion? Is it known?

Line 244: Function of the sas asRNA of phage P22 should also be described in more detail.

Line 269: „(like Rot)“ should be deleted

Line 275: verb „is“ is missing in sentence: „Esr41, a small RNA from…“

Line 339: The sentence „AntQ exerts strong cytotoxic effects on bacterial cells by interfering with transcription termination. This disruption leads to the accumulation of RNA polymerase on the DNA template, causing transcription-replication conflict.“ is repetition and should be omitted

Line 356: The sentence „The mechanisms by which IsrK influences anrP gene expression and its function in translational coupling were investigated [42].“ should be omitted

Author Response

Response to the Reviewer

Dear Reviewer,

Thank you very much for your valuable and thoughtful feedback. We truly appreciate your insights and have carefully incorporated your suggestions into our revised manuscript. All fragments changed by us in the manuscript are marked in green and, for convenience, also quoted below.

COMMENT 1 - Line 225: The sentence „It has been shown that DicF represses xylR and pykA mRNAs, which are involved in xylose metabolism and pyruvate kinase activity, respectively [22].“ should be moved into the line 209.

RESPONSE 1 - The sentence has been moved, following the suggestion of the reviewer, and now it is in lines 204-206.

COMMENT 2 - Line 240: What is the mechanism by which STnc6030 contributes to superinfection exclusion? Is it known?

RESPONSE 2 - Yes it is known. The text was extended and supplemented. Lines 230-245

“STnc6030 is a phage-derived antisense RNA (asRNA) transcribed from the BTP1 prophage genome and has been proposed to regulate gene expression by base-pairing with complementary mRNA sequences. Specifically, STnc6030 interacts with the mRNA of the BTP1 tailspike gene, reducing transcript stability and inhibiting translation [27]. This post-transcriptional regulation not only modulates expression of the tailspike protein - a key factor in host recognition and DNA injection - but also plays a pivotal role in superinfection exclusion, a mechanism that enables lysogenized bacteria to resist reinfection by the same or closely related phages. This exclusion contributes to the maintenance of the prophage state by preventing excessive phage replication and promoting the genomic stability of the host. Supporting this model, escape mutants of BTP1 were identified that could overcome the exclusion conferred by STnc6030. Sequencing of these variants revealed single nucleotide polymorphisms clustered in the region antisense to the tailspike gene, highlighting the importance of specific base-pairing interactions in the regulatory mechanism [27]. Notably, while STnc6030 limits superinfection, it does not impair induction of the BTP1 prophage, suggesting a fine-tuned regulatory system that selectively restricts external infection without interfering with prophage activation [27].”

COMMENT 3 - Line 244: Function of the sas asRNA of phage P22 should also be described in more detail.

RESPONSE 3 - Done. The text was extended and supplemented. Lines 249-260

“The mechanism by which sas contributes to SIE involves its function as an asRNA that regulates the expression of the sieB gene, which encodes the SieB exclusion protein, that is known to inhibit lytic development of superinfecting phages such as P22 [28]. asRNA sas exerts its effect through base-pairing interactions with sieB mRNA, modulating its translation. Disruption of this RNA-RNA interaction, for instance by targeted mutations, leads to altered sieB expression and consequently affects superinfection efficiency. This suggests that precise base-pairing between sas and its target mRNA is a key element of the exclusion mechanism [28]. Additionally, the system displays functional redundancy: even if sieB regulation is impaired, the Esc protein, which can be expressed by either the resident prophage or the incoming phage, can help bypass SieB-mediated exclusion. This indicates a complex regulatory balance between sas, SieB, and Esc.”

COMMENT 4 - Line 269: „(like Rot)“ should be deleted

RESPONSE 4 - It has been deleted. Line 282

COMMENT 5 - Line 275: verb „is“ is missing in sentence: „Esr41, a small RNA from…“

RESPONSE 5 - Added. Line 335

COMMENT 6 - Line 339: The sentence „AntQ exerts strong cytotoxic effects on bacterial cells by interfering with transcription termination. This disruption leads to the accumulation of RNA polymerase on the DNA template, causing transcription-replication conflict.“ is repetition and should be omitted

RESPONSE 6 - Omitted as requested. Line 360

COMMENT 7 - Line 356: The sentence „The mechanisms by which IsrK influences anrP gene expression and its function in translational coupling were investigated [42].“ should be omitted

RESPONSE 7 - Omitted as requested. Line 375